# High Incidence of Invasive Fungal Diseases in Patients with FLT3-Mutated AML Treated with Midostaurin: Results of a Multicenter Observational SEIFEM Study

**DOI:** 10.3390/jof8060583

**Published:** 2022-05-29

**Authors:** Chiara Cattaneo, Francesco Marchesi, Irene Terrenato, Valentina Bonuomo, Nicola Stefano Fracchiolla, Mario Delia, Marianna Criscuolo, Anna Candoni, Lucia Prezioso, Davide Facchinelli, Crescenza Pasciolla, Maria Ilaria Del Principe, Michelina Dargenio, Caterina Buquicchio, Maria Enza Mitra, Francesca Farina, Erika Borlenghi, Gianpaolo Nadali, Vito Pier Gagliardi, Luana Fianchi, Mariarita Sciumè, Pierantonio Menna, Alessandro Busca, Giuseppe Rossi, Livio Pagano

**Affiliations:** 1Hematology, Azienda Socio Sanitaria Territoriale-Spedali Civili, 25123 Brescia, Italy; erika.borlenghi@gmail.com (E.B.); giuseppe.rossi@asst-spedalicivili.it (G.R.); 2Hematology and Stem Cell Transplantation Unit, Istituto di Ricovero e Cura a Carattere Scientifico Regina Elena National Cancer Institute, 00144 Roma, Italy; francesco.marchesi@ifo.it; 3Unità Operativa Semplice Dipartimentale, Clinical Trial Center e Biostatistica e Bioinformatica, Istituto di Ricovero e Cura a Carattere Scientifico Regina Elena National Cancer Institute, 00144 Roma, Italy; irene.terrenato@ifo.it; 4Hematology Unit, Azienda Ospedaliera Universitaria Integrata, 37126 Verona, Italy; valentina.bonuomo1991@gmail.com (V.B.); gianpaolo.nadali@univr.it (G.N.); 5Hematology Unit, Fondazione Istituto di Ricovero e Cura a Carattere Scientifico Ca’ Granda Ospedale Maggiore Policlinico, 20122 Milan, Italy; nicola.fracchiolla@policlinico.mi.it (N.S.F.); mariarita.sciume@policlinico.mi.it (M.S.); 6Hematology and Stem Cell Transplantation Unit, Azienda Ospedaliero Universitaria Consorziale Policlinico, 70124 Bari, Italy; mario.delia74@gmail.com (M.D.); vitopier86@gmail.com (V.P.G.); 7Institute of Hematology, Università Cattolica del Sacro Cuore, 20123 Roma, Italy; marianna.criscuolo@policlinicogemelli.it (M.C.); luana.fianchi@policlinicogemelli.it (L.F.); livio.pagano@unicatt.it (L.P.); 8Division of Hematology and Stem Cell Transplantation, University Hospital of Udine, 33100 Udine, Italy; anna.candoni@asufc.sanita.fvg.it; 9Hematology and Stem Cell Transplant Unit, Ospedale Maggiore, 20122 Parma, Italy; lprezioso@ao.pr.it; 10Hematology, Ospedale San Bortolo, 36100 Vicenza, Italy; davide.facchinelli@aulss8.veneto.it; 11Haematology Unit, IRCCS Istituto Tumori “Giovanni Paolo II”, 70124 Bari, Italy; crescenza.pasciolla@gmail.com; 12Department of BioMedicine and Prevention, Tor Vergata University of Rome, 00133 Roma, Italy; del.principe@med.uniroma2.it; 13Unità Operativa di Ematologia e Trapianto di Cellule Staminali Emopoietiche Vito Fazzi, 73100 Lecce, Italy; miviforina@tiscali.it; 14Haematology and Bone Marrow Transplant Unit, Ospedale Monsignor R. Dimiccoli, 70051 Barletta, Italy; caterinabuquicchio@libero.it; 15Hematology, Policlinico Universitario “Paolo Giaccone”, 90127 Palermo, Italy; memitra@yahoo.com; 16Istituto di Ricovero e Cura a Carattere Scientifico, Ospedale San Raffaele, University Vita-Salute San Raffaele, 20132 Milan, Italy; farina.francesca@hsr.it; 17Department of Sciences and Technologies for Humans and the Environment, University Campus Bio-Medico of Rome, Italy, University Hospital Foundation Campus Bio-Medico of Rome, 00128 Roma, Italy; p.menna@policlinicocampus.it; 18Stem Cell Transplant Center, Azienda Ospedaliero-Universitaria Città della Salute e Della Scienza, 10126 Torino, Italy; abusca@cittadellasalute.to.it

**Keywords:** invasive fungal disease, acute myeloid leukemia, midostaurin, antifungal prophylaxis

## Abstract

The potential drug-drug interactions of midostaurin may impact the choice of antifungal (AF) prophylaxis in FLT3-positive acute myeloid leukemia (AML) patients. To evaluate the incidence of invasive fungal diseases (IFD) during the treatment of FLT3-mutated AML patients and to correlate it to the different AF prophylaxis strategies, we planned a multicenter observational study involving 15 SEIFEM centers. One hundred fourteen patients treated with chemotherapy + midostaurin as induction/reinduction, consolidation or both were enrolled. During induction, the incidence of probable/proven and possible IFD was 10.5% and 9.7%, respectively; no statistically significant difference was observed according to the different AF strategy adopted. The median duration of neutropenia was similar in patients with or without IFD. Proven/probable and possible IFD incidence was 2.4% and 1.8%, respectively, during consolidation. Age was the only risk factor for IFD (OR, 95% CI, 1.10 [1.03–1.19]) and complete remission achievement after first induction the only one for survival (OR, 95% CI, 5.12 [1.93–13.60]). The rate of midostaurin discontinuation was similar across different AF strategies. The IFD attributable mortality during induction was 8.3%. In conclusion, the 20.2% overall incidence of IFD occurring in FLT3-mutated AML during induction with chemotherapy + midostaurin, regardless of AF strategy type, was noteworthy, and merits further study, particularly in elderly patients.

## 1. Introduction

The standard treatment of acute myeloid leukemia (AML) has been chemotherapy according to the 3 + 7 scheme (daunorubicin 60 mg/sqm on days 1–3 as short i.v. infusion plus cytarabine 200 mg/sqm on days 1–7 as continuous i.v. infusion) for many decades [1,2]. Recently, new therapeutic scenarios have arisen in AML treatment, with the use of some small, oral molecule inhibitors, alone or in combination with classical chemotherapy or HMAs, such as tyrosine Fms-like tyrosine kinase 3 (FLT3) inhibitors. Midostaurin, a first-generation FLT3-inhibitor, added to induction and consolidation chemotherapy and as single agent for remission maintenance, significantly improved the overall survival and event-free survival of patients with FLT3-mutated AML in the RATIFY trial [3].

Midostaurin is a substrate of cytochrome P450 3A4 (CYP3A4), which converts midostaurin into O-demethylated or hydroxylated metabolites (CGP62221 or CGP52421, respectively) [4,5]. Since many other drugs are CYP3A4 inducers or inhibitors, drug-drug interactions (DDIs) of FLT3-inhibitors may be more complex than those occurring with classical chemotherapies. The azole antifungals, a milestone in both the prophylaxis and treatment of invasive fungal diseases (IFD), are CYP3A4 inhibitors and, therefore, may be responsible for an increase in midostaurin plasma levels.

Although a reduction in related mortality has been described, particularly concerning aspergillosis [6], IFDs are still a major infectious complication in AML patients, with a negative impact on outcome [7,8]. Posaconazole reduces the incidence of aspergillosis when used as antifungal (AF) prophylaxis during AML induction [9]. Because of its potent CYP3A4 inhibition activity, in patients treated with midostaurin some hematologists might prefer to use alternative prophylactic agents to avoid potential harmful DDIs.

No controlled data are available to guide the choice of AF drugs in this subset of patients. and only limited real-life data about IFD in FLT3-mutated AML treated with chemotherapy plus midostaurin have been reported. Therefore, we planned an observational study within the SEIFEM (Sorveglianza Epidemiologica InFezioni nelle EMopatie) group to document the incidence of IFDs according to the different AF prophylaxis strategies adopted in this setting.

## 2. Patients and Methods

### 2.1. Study Design

The present observational retrospective/prospective real-life study was conducted from 1 June 2019 to 31 December 2021 at 15 hematology units of tertiary care centers or university hospitals located throughout Italy and participating in SEIFEM.

The primary objective was to document the incidence of possible/probable/proven IFD occurring during induction chemotherapy in FLT3-mutated AML patients treated with 3 + 7 + midostaurin. The secondary objectives were to document: (1) the incidence of possible/probable/proven aspergillosis during induction chemotherapy in FLT3-mutated AML patients treated with 3 + 7 + midostaurin; (2) the incidence of possible/probable/proven IFDs during consolidation chemotherapy in FLT3-mutated AML patients treated with high dose Ara-C plus midostaurin; (3) the different AF prophylaxis strategies adopted according to the physician’s choice during induction chemotherapy in different Hematologic Centers and their correlation with IFD incidence.

The data were entered into case report forms. The impact of the age, gender, phase of treatment, type of AF prophylaxis adopted, ELN (European Leukemia Net) classification, type of FLT3 mutation and response to treatment on IFD occurrence was evaluated.

The study was conducted according to the guidelines of the Declaration of Helsinki and approved by the Institutional Review Board (or Ethics Committee) of ASST-Spedali Civili di Brescia (protocol code NP 4363, date of approval: 23 September 2020). The Ethics Committee of each participating site approved the use of the SEIFEM registry. Informed consent was obtained from all subjects involved in the study.

### 2.2. Definitions

IFDs were diagnosed according to EORTC/MSG criteria and categorized as possible, probable and proven [10].

The prognostic classification and response criteria of FLT3-mutated AML were made according to ELN 2017 criteria [11].

The duration of neutropenia was defined as the number of days with a neutrophil count below 500/mcL. Adverse events (AEs) were registered and graded according to the National Cancer Institute Common Toxicity Criteria (CTC), version 6.0. (https://ctep.cancer.gov/protocoldevelopment/electronic_applications/ctc.htm, accessed on 31 December 2021).

## 3. Statistical Analysis

Descriptive statistics were calculated for all variables of interest. Categorical variables were summarized through frequencies and percentage values while continuous variables through median values and their relative range. Each distribution was tested for normality with the Shapiro–Wilk test. Comparisons between groups were tested by the Mann–Whitney nonparametric test.

Overall Survival (OS) analyses were carried out by the Kaplan–Meier product-limit method and by Cox proportional hazard regression models. OS was defined as the time from the first diagnosis to the death of the patient or, if censored, last contact with the patient. 

The log-rank test was used to prove if any statistically significant difference between subgroups exists. 

The hazard risks and their relative 95% confidence intervals (95% CI) were estimated for each variable using the Cox univariate model and by adopting the most suitable prognostic modality as the referent group. Logistic regression models were conducted to individuate potential parameters involved in IFI onset. Multivariate models were then conducted by considering the variables significant at univariate analysis using the enter method. A *p*-value < 0.05 was considered statistically significant. Statistical analyses were carried out using SPSS software (SPSS version 21, SPSS Inc., Chicago, IL, USA). 

## 4. Results

### 4.1. Characteristics of Patients

Overall, 114 patients with FLT3-mutated AML treated with chemotherapy + midostaurin as induction/reinduction, consolidation or both were enrolled. All the patients received induction chemotherapy according to the “3 + 7” schedule plus midostaurin; in five cases, the duration was reduced (“2 + 5” schedule) because of advanced age. Table 1 summarizes the characteristics of enrolled patients.

The female gender was predominant (57%) and the median age was 55 years (range 18–78). FLT3 ITD and TKD mutation were present in 94 (82.5%) and 20 (17.5%) patients, respectively; in 70 (61.4%), NPM1 mutation was also detected. AF prophylaxis was delivered during induction to 106/114 patients (93%); AF strategy was heterogeneous and differed among centers. Posaconazole was used in 55 patients (48.2%) and echinocandins (micafungin or caspofungin) in 18 (15.8%). In 24 patients (21%), a sequential AF strategy was adopted, consisting of administration of posaconazole in the first 7 days of chemotherapy followed by echinocandin (micafungin or caspofungin) during midostaurin treatment. Overall, 79 patients received a posaconazole-based prophylaxis. A dose reduction of posaconazole was recorded in only one case (200 mg/d). Midostaurin was reduced (50% of the dosage) in two cases during posaconazole prophylaxis; both the patients achieved a complete remission (CR).

AF prophylaxis was also delivered in the 12 patients undergoing reinduction (100%) and in 73/160 consolidation courses (45.6%). The median duration of neutropenia during induction chemotherapy was 22 days (8–180 days); we did not observe any statistical difference between patients receiving posaconazole during midostaurin administration (21 days, range 11–180) and all other patients (23 days, range 8–59) (*p* = 0.658). Overall, 72.8% of patients achieved CR after the first induction course.

### 4.2. Incidence of IFD

Overall, 34 IFD were recorded throughout all treatment phases: 17 possible, 12 probable and 5 proven (Figure 1).

During induction chemotherapy, IFD incidence was 23/114 (20.2%) when considering also possible IFD; it was 12/114 (10.5%) when considering only probable/proven IFD. The median time from the induction chemotherapy start and possible and probable/proven IFD was 16.5 (range 6–53) and 14 (range 2–32) days, respectively. The lung was involved in all cases of possible and probable IFD; in only one case, concomitant mycotic pneumonia and sinusitis (aspergillosis) was observed. *Aspergillus* spp. and *Saprochaete capitata* were responsible for seven probable and one proven IFD, respectively, whereas *Candida* spp. were responsible for the four proven IFD (candidemia in all cases, due to *C. grabrata*, *C. krusei*, *C. krusei* + *C. incospicua*, and *Candida* spp.).

Thirty-one of one hundred fourteen (27.2%) patients did not achieve CR after first induction and twelve underwent reinduction with chemo + midostaurin. Four IFD were observed (33.3%, three possible and one probable aspergillosis).

Consolidation chemotherapy + midostaurin was administered in 79 patients, for a total of 160 courses. Seven cases of IFD were recorded (4.4%); all but one occurred during the first consolidation course. They were possible and probable aspergillosis in three cases each, and candidemia (*C. parapsilosis*) in one case. Table 2 summarizes the characteristics of proven/probable IFD.

### 4.3. Association between IFD and AF Prophylaxis Strategy

Two out of the eight patients (25%) who did not receive AF prophylaxis developed a possible IFD during induction chemotherapy. Twenty-one patients on any AF prophylaxis developed IFD (19.8%). No significant differences in terms of IFD incidence were observed in patients who received different AF strategies during induction chemotherapy, although IFD was observed more frequently in patients receiving an echinocandin alone as AF strategy (5/18, 27.7%%), compared to posaconazole (10/55, 18.1%) or the posaconazole → echinocandin sequential strategy (3/24, 12.5%) (*p* = 0.5 and *p* = 0.26, respectively). Figures were similar if we consider only proven/probable IFD (echinocandins: 3/18, 16.7%; posaconazole 5/55; 9.1%, posaconazole → echinocandin 2/24, 8.3%) (Figure 2).

Probable aspergillosis was observed in 1/18 (5.5%) patients who received echinocandins alone, compared to 5/79 (6.3%) treated with posaconazole (4/55, 7.3%, posaconazole alone strategy and 1/24, 4.2% sequential posaconazole → echinocandins strategy). 

The four IFD observed during reinduction occurred during posaconazole (two possible IFD) and during echinocandin (one possible IFD and one probable aspergillosis) as AF prophylaxis.

All seven IFD observed during consolidation treatment were observed in patients who did not receive AF prophylaxis at all (two possible IFD, three probable aspergillosis and one *C. parapsilosis* fungemia) or who received fluconazole (one possible IFD). 

### 4.4. Risk Factors for IFD

During induction chemotherapy, age was the only risk factor for IFD occurrence, particularly for proven/probable IFD (Table 3a,b). The median duration of neutropenia was similar in patients with or without an IFD diagnosis (IFD: 22 days (13–75) vs. no IFD: 22 days (8–180)) (*p* = 0.810). In patients who did not receive a posaconazole prophylaxis, the median duration of neutropenia was similar in IFD vs. no IFD patients: 26 (13–59) vs. 23 (8–46) (*p* = 0.714)

No predictive factors for IFD during consolidation were detected. However, the IFD incidence in patients who did not receive a mold-active AF prophylaxis was 7.1% (7/99) compared to 0/68 in patients who received an anti-mold AF prophylaxis (*p* = 0.042).

### 4.5. Midostaurin Discontinuation

Midostaurin was discontinued during induction in 17 patients; in 10 cases because of infections, including 5 breakthrough IFD, 4 neutropenic enterocolitis and 1 FUO, and in 7 cases (6.1%) because of grade 3–4 toxicity (QTc prolongation in 4, gastrointestinal toxicity in 2 and abnormal liver function tests in 1). Furthermore, 7 out of the 16 patients were receiving posaconazole at midostaurin stop, with a discontinuation rate of 7/55 (12.7%), as compared to no posaconazole prophylaxis (9/59, 15.3%). Toxicity was responsible for midostaurin discontinuation in three (5.5%) cases (one QTc prolongation, one abnormal liver function test and one gastrointestinal toxicity) during posaconazole prophylaxis and in four (6.8%, three QTc proplongation and one gastrointestinal toxicity) in all other cases.

During consolidation, midostaurin was discontinued in two cases: one during the first course because of a septic shock during caspofungin prophylaxis and the other during the second course because of a severe headache during posaconazole prophylaxis. No cases of midostaurin discontinuation were recorded during the 12 reinduction courses.

### 4.6. Outcome

After a median follow-up of 5 months (range: 1–32), 18 patients were deceased. Thirty-day mortality was 2/12 for patients with probable/proven IFD and 0/11 for those with possible IFD. In only one case of candidemia, IFD was responsible for death during induction chemotherapy. Therefore, considering only proven/probable IFD, the attributable mortality during induction was 8.3% (1/12). In another case, probable aspergillosis was a contributing cause of death together with refractory leukemia.

As expected, OS was superior in patients achieving CR after first induction than in those with a refractory disease (median OS: 22 months, 95% CI: 13.6–30.4 vs. 11 months 95% CI: 7.3–14.7 in patients with a CR vs. refractory, respectively, *p* < 0.001). The response to AML treatment and the duration of neutropenia were the only predictive factors of survival (Table 4). IFD occurrence did not impact on the survival probability of patients undergoing induction chemotherapy (median OS: 22 months, 95% CI: 0–44.5 vs. 16 months, 95% CI: 9.4–22.6 in patients with and without IFD, respectively, *p* = 0.221) (Figure 3a), but it negatively affected survival in patients who did not achieve a CR after first induction chemotherapy (median OS: 8 months, 95% CI: not estimable vs. 11 months, 95% CI: 6.3–15.2 in refractory patients with and without IFD, respectively, *p* = 0.026) (Figure 3b).

## 5. Discussion

The recent availability of new drugs has modified the therapeutic management of AML. However, the possibility of interactions in the metabolism of different drug classes may be responsible for altered plasma levels and consequent toxicity or reduced activity of the involved therapeutic agents. An unresolved issue is the DDI between FLT3-inhibitors and azoles. Indeed, an increased midostaurin toxicity has been reported in elderly patients concomitantly treated with azole AF prophylaxis [12], whereas no serious adverse events were reported in patients up to 60 years in the RATIFY trial, where the use of posaconazole was allowed. These considerations are responsible for the concerns of hematologists about the use of AF prophylaxis and about the choice among available antifungal agents, with potential consequences on infectious events or toxicity.

The present study is the largest study investigating the AF policies adopted by hematologists in a real-life setting when treating AML patients with FLT3-inhibitors. It confirms the heterogeneity of the AF strategies utilized, highlighting the difficulties encountered by clinicians and the lack of firm evidence supporting their choice. Posaconazole, the standard AF agent used in the prophylaxis of AML, was used during midostaurin administration in only 48% of patients during induction. Echinocandins were the most frequent alternative to posaconazole, and they were also utilized in a sequential schedule in 24% of patients, with the administration of posaconazole for the first 7 days, followed by echinocandins at the introduction of midostaurin. In addition, AF agents not registered for use as prophylaxis, such as caspofungin, L-AmB or isavuconazole, were also used in several cases. We did not observe any statistically significant difference in IFD incidence according to the different AF prophylaxis, although the choice of using echinocandins only during the induction period was associated with the highest IFD incidence (27.7%), as compared to the lowest IFD rate (12.5%) during the sequential (posaconazole → echinocandins) strategy.

The incidence of IFD (19.8%) in patients receiving AF prophylaxis was higher than expected during induction chemotherapy, regardless of the kind of AF prophylaxis adopted; it was even similar to that observed in the eight patients who did not receive AF prophylaxis at all (25%). In a small (*n* = 22) cohort of AML patients treated with “3 + 7 + midostaurin”, Phoompoung et al. [13] reported an incidence of proven/probable invasive mold infections similar to patients treated with classical “3 + 7” (4.3% vs. 4.5%). In our study, the proven/probable IFD incidence was 9.1% in patients receiving posaconazole prophylaxis during midostaurin treatment. Similarly, the aspergillosis incidence was high (7.3%) in patients treated with posaconazole. The IFD incidence in our series was confirmed as higher when compared to that observed in AML patients treated with the 3 + 7 scheme without midostaurin in the same period of observation (5.5%). IFD diagnosis was correlated to advanced age, but age did not differ across patient groups receiving different AF prophylaxis. In our study on FLT3-mutated AML, the median age in the posaconazole group (53.5 years) was similar to that reported by Cornely et al.^9^ in unselected patients affected by AML who demonstrated a IFD proven/probable IFD incidence of 2% in the posaconazole arm. The high IFD incidence detected in our study needs to be investigated.

FLT-3 mutation may confer an intrinsic predisposition to IFD or resistance to AF prophylaxis. Among 12 proven/probable IFD observed during induction, 4 were candidemia, and 3 were detected in patients receiving echinocandins as the AF agent of choice for candidemia. However, scarce data in the literature support such a hypothesis. It is known that FLT3 is an immune-enhancing molecule, but data concerning the effect of FLT3 mutation on the immune response are lacking. The lower rate of CR expected in the more aggressive FLT3-mutated AML may also predispose one to IFD [14,15] but in our study, 72.8% of patients treated with chemotherapy + midostaurin achieved a CR after first induction, as opposed to 59% in the midostaurin arm of the RATIFY study. 

Alternatively, treatment with FLT-3 inhibitors may be a predisposing factor. In the RATIFY study [3], for enrolling patients aged less than 60, no differences were observed in terms of infections in the midostaurin group compared to the placebo group, despite a longer duration of neutropenia (median 26 days vs. 21 days). However, in the RATIFY study, no specific data about IFD were provided, and data on the incidence of IFD in large cohorts of patients treated with midostaurin are still lacking. The well documented immunomodulating effects of FLT3 inhibitors may support our data. Indeed, in vitro and in vivo studies showed that their use can result in the inhibition of type I IFN production and in impairment of dendritic cells’ (DC) development [16,17,18], two important players involved in the host’s immune response against infections, including IFD. FLT3 expression on progenitor cells is necessary for the development of DCs, which play an essential role in immunity as they serve as a link between the innate and adaptive immune system and have an extremely potent capacity to activate naive T cells.

Despite its limited efficacy during induction, AF prophylaxis was beneficial during consolidation. IFDs were observed only in patients who did not receive anti-mold prophylaxis (overall: 8%, proven/probable IFD: 4.6%), and probable aspergillosis accounted for 3.4%, regardless of age. The incidence was similar to that observed in the SEIFEM 2016 study in the group of patients without AF prophylaxis [19]. 

From our data, we confirmed a low IFD-related mortality, as already reported [20]. However, a negative impact of IFD on outcome was observed in AML patients not in CR after first induction, as survival was worse in refractory patients with IFD compared to those without. Therefore, IFD occurrence is confirmed to be a predictive factor for dismal prognosis, as reported by other authors [7,14,21].

Toxicity was not different to that expected [3,12]. In addition, the choice of posaconazole as AF prophylaxis during midostaurin treatment did not translate to a higher rate of midostaurin discontinuation, as only 5.5% of patients developed toxicity potentially correlated with increased midostaurin plasma levels, as compared to 6.8% in patients not receiving posaconazole. Our observation indirectly confirms data reported in the post hoc analysis of the RATIFY trial [3], where, despite a 1.44-fold increase in midostaurin exposure, no increase in midostaurin-related adverse events were observed in patients who concomitantly received strong CYP3A4 inhibitors [22]. However, the high number of cardiac adverse events (up to 13% grade 3–4) observed in the German-Austrian AML Study Group 16–10 [12] in this category of patients when treated with both midostaurin and strong CYP3A4 inhibitors confirms the urgent need of pharmacokinetic studies. Indeed, midostaurin levels in patients who concomitantly received posaconazole were found to be increased up to ≥ 8-fold in the study of Menna et al. [23].

Our study has some limitations. First, as the main objective of our study was the IFD incidence in patients treated with chemotherapy plus midostaurin, data concerning the neutrophil and lymphocyte count at IFD were not requested and, therefore, we could not evaluate the impact of these variables on the risk of IFD development. Moreover, as we analyzed the IFD incidence induction and consolidation separately, the impact of disease status on risk of IFD was not evaluated.

In conclusion, IFD incidence is higher than expected in patients treated with 3 + 7 + midostaurin, regardless of posaconazole, echinocandins or other antifungal prophylactic strategies. More severe toxicity potentially correlated to higher midostaurin plasma levels in patients simultaneously treated with strong CYP3A4 inhibitors was not detected. A pharmacokinetic study is ongoing within the SEIFEM group to better clarify the impact of coadministration of strong CYP3A4 inhibitors and midostaurin, particularly in the older population.

## Figures and Tables

**Figure 1 jof-08-00583-f001:**
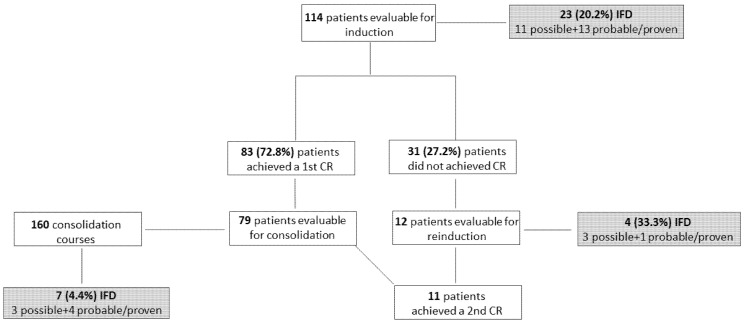
Flowchart of the enrolled patients. IFD: invasive fungal disease; CR: complete remission.

**Figure 2 jof-08-00583-f002:**
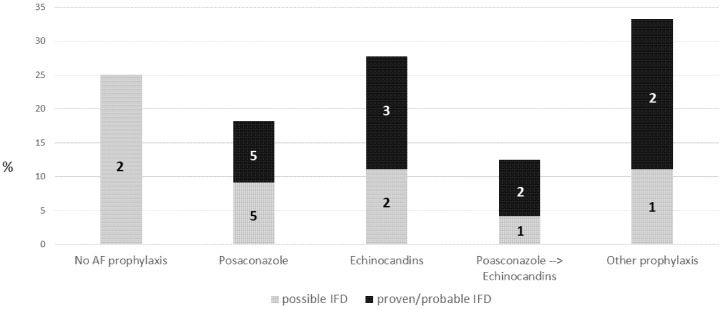
IFD incidence during induction chemotherapy according to different AF prophylaxis. IFD: invasive fungal disease.

**Figure 3 jof-08-00583-f003:**
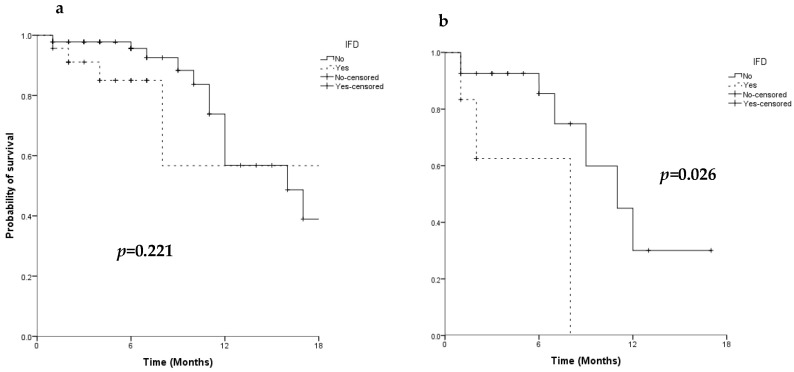
Overall survival of 114 patients (**a**) and of 31 primary refractory patients (**b**) according to IFD occurrence.

**Table 1 jof-08-00583-t001:** Characteristics of 114 enrolled patients.

**M/F ratio**	49/65
**Median age, years (range)**	55 (18–78)
**FLT3 ITD/TKD mutation (%)**	94 (82.5)/20(17.5)
**NPM-1 mutation (%)**	70 (61.4)
**ELN (low/not low) (%)**	34 (32.7)/70 (67.3) *
**AF prophylaxis (induction cht, *n* = 114) (%)** No Fluconazole Posaconazole Posaconazole → Echinocandin Echinocandin L-AmB Isavuconazole	8 (7) 3 (2.6) 55 (48.2) 24 (21.1) 18 (15.8) 4 (3.5) 2 (1.8)
**AF prophylaxis (reinduction cht, *n* = 12) (%)** No Posaconazole Posaconazole → Echinocandin Echinocandin	2 (1.7) 3 (25) 1 (7.3) 6 (50)
**AF prophylaxis (consolidation cht, *n* = 160)** No Fluconazole Posaconazole Posaconazole → Echinocandin Echinocandin L-AmB Isavuconazole Voriconazole	87 (54.3) 8 (5) 20 (12.5) 6 (3.8) 27 (16.9) 1 (0.6) 6 (3.8) 5 (3.1)
**Median duration of PMN<500/mmc (days) ****	22 (range 8–180)
**Midostaurin suspension (induction) (%)**	16/114 (14)
**CR after first induction (%)**	83 (72.8%)
**Median follow-up, months (range)**	5 (1–32)

* ELN risk class evaluable in 104 patients; ** evaluable in 105 patients. ELN: European Leukemia Net; AF: antifungal; cht: chemotherapy; L-AmB: liposomal amphotericin B; PMN: polymorphonucleated; CR: complete remission.

**Table 2 jof-08-00583-t002:** Characteristics of proven/probable IFD.

	Type of IFD	Microbiology/Biomarker
**Induction** **(active hematologic disease)**		
**1**	Pulmonary aspergillosis	GM serum
**2**	Pulmonary aspergillosis	GM BAL
**3**	Pulmonary + sinus aspergillosis	GM serum
**4**	Pulmonary aspergillosis	*A. terreus* (BAL)
**5**	Pulmonary aspergillosis	GM serum
**6**	Pulmonary aspergillosis	GM serum/BAL
**7**	Geothricosis	*S. capitata* (lung biopsy)
**8**	Pulmonary aspergillosis	GM serum
**9**	Candidemia	*Candida* spp.
**10**	Candidemia	*C. krusei*
**11**	Candidemia	*C. krusei* + *C. incospicua*
**12**	Candidemia	*C. glabrata*
**Reinduction** **(active hematologic disease)**		
**1**	Pulmonary aspergillosis	GM serum
**Consolidation** **(controlled hematologic disease)**		
**1**	Pulmonary aspergillosis	GM serum
**2**	Pulmonary + sinus aspergillosis	GM BAL
**3**	Pulmonary aspergillosis	GM serum
**4**	Candidemia	*C. parapsilosis*

IFD: invasive fungal disease; GM: galactomannan; BAL bronchoalveolar lavage.

**Table 3 jof-08-00583-t003:** (**a**) Risk of IFD (proven + probable vs possible + no) estimates for patients at 1st induction. (**b**) Risk of IFD (proven + probable + possible vs no) estimates for patients at 1st induction.

**(a)**			
**IFD = 12**		**Univariate**	
	**Comparison**	**OR (95% CI)**	** *p* ** **-Value**
**Age**	Continuous	**1.10 (1.03–1.19)**	**0.008**
**Gender**	Female vs. Male	0.73 (0.22–2.42)	0.605
**FLT3**	TKD vs. ITD	2.50 (0.68–9.25)	0.170
**NPM**	Pos vs. Neg	0.88 (0.26–2.97)	0.838
**ELN**	Not low vs. Low	0.97 (0.27–3.47)	0.960
**AF prophylaxis**	Yes vs. No	not estim	not estim
**Neutropenia (days) ***	Continuous	0.99 (0.95–1.03)	0.665
**Posaconazole containing prophylaxis**	Yes vs. No	0.58 (0.17–1.98)	0.388
**(b)**			
**IFD = 23**		**Univariate**	
	**Comparison**	**OR (95% CI)**	** *p* ** **-Value**
**Age**	Continuous	**1.06 (1.01–1.10)**	**0.021**
**Gender**	Female vs. Male	0.98 (0.39–2.45)	0.957
**FLT3**	TKD vs. ITD	1.30 (0.42–4.02)	0.647
**NPM**	Pos vs. Neg	0.79 (0.31–2.00)	0.617
**ELN**	Not low vs. Low	1.05 (0.38–2.88)	0.922
**AF prophylaxis**	Yes vs. No	0.74 (0.14–3.94)	0.725
**Neutropenia (days) ***	Continuous	0.99 (0.97–1.02)	0.861
**Posaconazole containing prophylaxis**	Yes vs. No	0.49 (0.19–1.27)	0.141

IFD: invasive fungal disease; ELN: European Leukemia Net; AF: antifungal. * evaluable in 105 patients

**Table 4 jof-08-00583-t004:** Predictive factors for survival (patients at 1st induction).

.		Univariate		Multivariate	
	Comparison	HR (95% CI)	*p*-Value	HR (95% CI)	*p*-Value
**Age**	continuous	1.03 (0.98–1.08)	0.222		
**Gender**	Female vs. Male	1.11 (0.40–3.02)	0.846		
**FLT3**	TKD vs. ITD	0.73 (0.17–3.22)	0.679		
**NPM**	Pos vs. Neg	1.02 (0.39–2.63)	0.974		
**ELN**	Not low vs. Low	1.53 (0.51–4.58)	0.452		
**AF prophylaxis**	Yes vs. No	0.63 (0.14–2.80)	0.539		
**Neutropenia (days) ***	continuous	**1.03 (1.01–1.05)**	**0.002**	**1.02 (1.00–1.04)**	**0.019**
**Posaconazole containing prophylaxis**	Yes vs. No	1.64 (0.61–4.41)	0.331		
**IFD**	Yes vs. No	1.90 (0.66–5.47)	0.235		
**possible IFD**	Yes vs. No	1.86 (0.22–15.67)	0.568		
**probable IFD**	Yes vs. No	1.57 (0.35–6.98)	0.554		
**proven IFD**	Yes vs. No	1.81 (0.40–8.15)	0.438		
**Response to AML tratment**	No CR vs. CR	**5.11 (1.92–13.60)**	**0.001**	**4.68 (1.47–14.92)**	**0.009**

IFD: invasive fungal disease; ELN: European Leukemia Net; AF: antifungal. * evaluable in 105 patients.

## Data Availability

Not applicable.

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
