# Peer review of "High Incidence of Invasive Fungal Diseases in Patients with FLT3-Mutated AML Treated with Midostaurin: Results of a Multicenter Observational SEIFEM Study"

_jof, 2022, doi:10.3390/jof8060583_

Round 1

Reviewer 1 Report

In their manuscript, C. Cattaneo et al performed a retrospective/prospective multi-center  observational study of IFD incidence of FLT3-mutated AML treated with midostaurin and anti-fungal policies adopted. The conclusions reached by the authors would be of fairly interest to the community of AML researchers. 

However, there is one further  improvement should be considered by authors. Overall, IFD incidence of AML treated with "3+7" scheme and midostaurin in this study is relatively high as mentioned in article tittle.  The real IFD incidence of AML population treated with standard  "3+7"scheme in these centers at the same period should be included in this study as the control group. 

A few spelling errors should be corrected. For example, Paragraph 50, the first appearance of "3+7" scheme should provide a brief full name or explanation. Paragraph 316, Ratify study should be RATIFY trial/study. 

Author Response

Response to Reviewer 1 Comments

Point 1: In their manuscript, C. Cattaneo et al performed a retrospective/prospective multi-center observational study of IFD incidence of FLT3-mutated AML treated with midostaurin and anti-fungal policies adopted. The conclusions reached by the authors would be of fairly interest to the community of AML researchers. 

However, there is one further improvement should be considered by authors. Overall, IFD incidence of AML treated with "3+7" scheme and midostaurin in this study is relatively high as mentioned in article tittle. The real IFD incidence of AML population treated with standard “3+7"scheme in these centers at the same period should be included in this study as the control group. 

Response 1: We thank the reviewer for this comment, which allows us to confirm the high incidence of IFD in AML patients treated with 3+7+midostaurin. However, as the comparison with AML patients tretated according to 3+7 scheme without midostaurin was not an objective of our study, we added a sentence in the “Discussion” (IFD incidence in our series was confirmed higher when compared to that observed in AML patients treated with 3+7 scheme without midostaurin in the same period of observation (5.5%)”)

Point 2: A few spelling errors should be corrected. For example, Paragraph 50, the first appearance of "3+7" scheme should provide a brief full name or explanation. Paragraph 316, Ratify study should be RATIFY trial/study. 

Response 2: We amended, as requested.

Reviewer 2 Report

Cattaneo and colleagues present the results of a multicenter observational study of the epidemiology of invasive fungal infections in AML patients receiving midostaurin. This topic is significant due to potential drug-drug interactions with antifungal prophylaxis and the so far poorly studied fungal infection landscape in this cohort. The results are unique and of general interest to the medical mycology community. However, the paper needs significant improvement of data presentation (tables and figures) and some additional analyses/details would be desirable, as specified below.

Specific comments:

  1. The abstract needs an introductory sentence highlighting the significance of the topic.
  2. If the goal of the study is to review “patients with FLT3-mutated AML treated with midostaurin” (title), then the 5 patients not receiving midostaurin should be excluded from all analyses.
  3. Line 120: The writing suggests that a (multivariate?) logistic regression model was performed; however, no such results are presented in the manuscript. Multivariate analysis would be an important addition to the manuscript, if feasible based on the number of data sets.
  4. The chemotherapy regimens (3+7 etc.) need to be introduced. Not all readers from the mycological community might be familiar with them.
  5. Tables 1-4 are all very superficial and lack many important risk factors, e.g., persistent neutropenia, etc. Table 2 should include even more granular information such as ANC/ALC at time of infection, remission status at time of infection, co-infections, etc.
  6. Lines 141-144: Addition of TDM data (for posaconazole and midostaurin) would enhance the manuscript and allow to better understand the relevance of drug-drug interactions in the authors’ cohort.
  7. Please provide percentages in addition to patient numbers in Table 1.
  8. Introduce all abbreviations used in the table in a footnote below the table.
  9. Figure 1 has a weird design and should be edited for clarity. I would suggest starting with the 119 patients, then exclude the 5 without midostaurin, then distinguish by CR status, then specify chemotherapy, and then mention numbers and percentages of patients with IFD per group.
  10. For Figures 1 and 2, please make the fonts larger and use a vector format. The figures are difficult to read when printed on letter page format.
  11. Line 216-219: It is unclear which risk factors were considered for the analyses presented – only the ones shown in the tables (which would be quite superficial, as stated above), or were additional factors considered?
  12. In order to better interpret the survival data shown in Figure 3, it would be important to include some data (or figure) in the paper indicating the time from first diagnosis to the development of IFD. It would also be helpful to split Figure 3 into 2 panels, one comparing all 114 midostaurin-treated patients with or without IFD, followed by the current panel showing those without CR.
  13. What was the 30- or 42-day mortality after IDF diagnosis in these patients?
  14. Line 290: The first part of this sentence needs to make clear that the 19.8% IFD rate refers to patients on antifungal prophylaxis.

Author Response

Response to Reviewer 2 Comments

Cattaneo and colleagues present the results of a multicenter observational study of the epidemiology of invasive fungal infections in AML patients receiving midostaurin. This topic is significant due to potential drug-drug interactions with antifungal prophylaxis and the so far poorly studied fungal infection landscape in this cohort. The results are unique and of general interest to the medical mycology community. However, the paper needs significant improvement of data presentation (tables and figures) and some additional analyses/details would be desirable, as specified below.

Point 1: The abstract needs an introductory sentence highlighting the significance of the topic.

Response 1: We added an introductory sentence in the abstract, as requested (“The potential drug-drug interactions of midostaurin may impact the choice of antifungal (AF) prophylaxis in FLT3-positive acute myeloid leukemia (AML) patients”). However, to stay within the 200 word count, as required, we have reduced some parts.

Point 2: If the goal of the study is to review “patients with FLT3-mutated AML treated with midostaurin” (title), then the 5 patients not receiving midostaurin should be excluded from all analyses.

Response 2: We excluded the 5 patients not receiving midostaurin during induction, as requested. Text, Table 1 and Figure 1 were amended accordingly.

Point 3: Line 120: The writing suggests that a (multivariate?) logistic regression model was performed; however, no such results are presented in the manuscript. Multivariate analysis would be an important addition to the manuscript, if feasible based on the number of data sets.

Response 3: We thank the Reviewer for arising this point but we did not conduct any multivariate analyisys neither in Logistic regresion models nor in Cox models because only one parameter resulted statistically significant in the univariate analysis: age at diagnosis in tabe 3a/3b and response to AML treatment in table 4. As the referee suggested, we added a new parameter, that is “days of neutropenia” as a potential risk factor in both models. Given the obtained results, we then conducted a multivariate Cox model as you can read in the new version of table 4. We modified the text in the manuscript accordingly.

Point 4: The chemotherapy regimens (3+7 etc.) need to be introduced. Not all readers from the mycological community might be familiar with them.

Response 4: an explanation of the 3+7 scheme has been provided in the introduction, as requested also by Reviewer 1

Point 5: Tables 1-4 are all very superficial and lack many important risk factors, e.g., persistent neutropenia, etc. Table 2 should include even more granular information such as ANC/ALC at time of infection, remission status at time of infection, co-infections, etc.

Response 5: We agree with the Reviewer and we added a new parameter, that is “days of neutropenia” as a potential risk factor in Tables 3a/3b and 4. We also added the remission status at IFD in Table 2. Unfortunately, we do not have the neutrofil and lymphocyte count at IFD, and we have emphasized it in the “Discussion”, as a limit of the study.

Point 6: Lines 141-144: Addition of TDM data (for posaconazole and midostaurin) would enhance the manuscript and allow to better understand the relevance of drug-drug interactions in the authors’ cohort.

Response 6: We agree with the Reviewer about TDM data for posaconazole and midostaurin. Indeed, a study concerning TDM of midostaurin is also ongoing within the SEIFEM, as stated at the end of the “Discussion” (“A pharmacokinetic study is ongoing within the SEIFEM group to better clarify the impact of coadministration of strong CYP3A4 inhibitors and midostaurin, particularly in the older population”). Unfortunately, data are not yet available.

Point 7: Please provide percentages in addition to patient numbers in Table 1.

Response 7: We added the percentages in addition to patient numbers in Table 1, as requested.

Point 8. Introduce all abbreviations used in the table in a footnote below the table.

Response 8: We added all the abbreviations used in the tables.

Point 9: Figure 1 has a weird design and should be edited for clarity. I would suggest starting with the 119 patients, then exclude the 5 without midostaurin, then distinguish by CR status, then specify chemotherapy, and then mention numbers and percentages of patients with IFD per group.

Response 9: We amended Figure 1, as requested.

Point 10: For Figures 1 and 2, please make the fonts larger and use a vector format. The figures are difficult to read when printed on letter page format.

Response 10: We made the fonts larger for Figures 1 and 2 and we used an high resolution format.

Point 11: Line 216-219: It is unclear which risk factors were considered for the analyses presented – only the ones shown in the tables (which would be quite superficial, as stated above), or were additional factors considered?

Response 11: As already reportd at point 3, we added days of neutropenia in tabs 3a,3b e 4.

Point 12: In order to better interpret the survival data shown in Figure 3, it would be important to include some data (or figure) in the paper indicating the time from first diagnosis to the development of IFD. It would also be helpful to split Figure 3 into 2 panels, one comparing all 114 midostaurin-treated patients with or without IFD, followed by the current panel showing those without CR.

Response 12: We agree with the referee about including data indicating time to IFD. We addded a sentence in the Results – Incidence of IFD (Median time from induction chemotherapy start and possible and probable/proven IFD was 16.5 (range 6-53) and 14 (range 2-32) days, respectively.”). Figure 3 was modified as suggested.

Point 13: What was the 30- or 42-day mortality after IDF diagnosis in these patients?

Response 13: 30-day mortality was 2/12 (16.7%) in probable/proven IFD and 0/11 possible IFD. We added a sentence in the Results Outcome (“Thirty-day mortality was 2/12 for patients with probable/proven IFD and 0/11 for those with possible IFD”)

Point 14: Line 290: The first part of this sentence needs to make clear that the 19.8% IFD rate refers to patients on antifungal prophylaxis.

Response 14: We amended the text specifying that it refers to patients receiving antifungal prophylaxis.

Round 2

Reviewer 1 Report

I have reviewed the revision. The authors successfully responded to my comments/suggestions, and thus, the manuscript can be accepted for publication in its current form.

Reviewer 2 Report

The authors were responsive to my comments and have addressed most points by adding data and/or acknowledging limitations. The quality of the display items is still suboptimal (table format, resolution and font size of the figures), but from my own experience publishing in JOF, I am aware that this will be addressed during editorial processing.